# *Drosophila* Myc restores immune homeostasis of Imd pathway via activating miR-277 to inhibit *imd/Tab2*

**Ruimin Li**, **Hongjian Zhou**, **Chaolong Jia**, **Ping Jin**\*, **Fei Ma**\*

Laboratory for Comparative Genomics and Bioinformatics & Jiangsu Key Laboratory for Biodiversity and Biotechnology, College of Life Science, Nanjing Normal University, Nanjing, China

\* jinping8312@163.com (PJ); mafei01@tsinghua.org.cn (FM)

**Data Availability Statement:** All relevant data are within the manuscript and its Supporting Information files.

**Funding:** This work was supported by grants from the National Natural Science Foundation of China

## Abstract

*Drosophila* Myc (dMyc), as a broad-spectrum transcription factor, can regulate the expression of a large number of genes to control diverse cellular processes, such as cell cycle progression, cell growth, proliferation and apoptosis. However, it remains largely unknown about whether dMyc can be involved in *Drosophila* innate immune response. Here, we have identified dMyc to be a negative regulator of *Drosophila* Imd pathway via the loss- and gain-of-function screening. We demonstrate that dMyc inhibits *Drosophila* Imd immune response via directly activating *miR-277* transcription, which further inhibit the expression of *imd* and *Tab2-Ra/b*. Importantly, dMyc can improve the survival of flies upon infection, suggesting inhibiting *Drosophila* Imd pathway by dMyc is vital to restore immune homeostasis that is essential for survival. Taken together, our study not only reports a new dMyc-miR-277-imd/Tab2 axis involved in the negative regulation of *Drosophila* Imd pathway, and provides a new insight into the complex regulatory mechanism of *Drosophila* innate immune homeostasis maintenance.

## Author summary

Innate immunity is the first line of defense against pathogenic microorganisms. Both overactivation and depression of immune response are detrimental to the organism. It is indispensable to regulate the duration and intensity of immune response. In this work, we find that *Drosophila* Myc (dMyc) as a transcription factor activates the transcription of miR-277 to negatively regulate the Imd pathway via inhibiting the expression of *imd/Tab2* gene in the middle and later stage of *Drosophila* innate immune. dMyc is required to restore *Drosophila* immune homeostasis for the survival of flies upon infection. Since dMyc is well conserved in animals, our findings will be important in understanding the complex regulatory mechanisms of innate immune responses in animals.

under Grant No.31970477 (FM), No.31572324 (FM) and a Project Funded by the Priority Academic Program Development of Jiangsu Higher Education Institutions. The funders had no role in study design, data collection and analysis, decision to publish, or preparation of the manuscript.

**Competing interests:** The authors have declared that no competing interests exist.

# Introduction

Innate immune system plays critical roles in host defensing foreign pathogenic microorganisms [1]. *Drosophila melanogaster* is an important model for studying innate immune response in animals. *Drosophila* involves both cellular and humoral mechanisms to produce diverse antimicrobial peptides (AMPs) to resist the invasion of foreign pathogens via innate immune responses [2, 3]. Transcriptional expressions of *Drosophila AMPs* are primarily controlled by Toll and the immune deficiency (Imd) signaling pathways [4, 5]. *Drosophila* mainly utilizes the Imd signaling pathway to resist Gram-negative bacteria infection [6]. Currently, although the activation mechanisms of innate immune response have been well-established, the study on restoration mechanism of innate immune homeostasis remains a major challenge [7].

The intensity and duration of *Drosophila* immune response can be positively or negatively regulated at multiple layers [8]. For example, STING and sick can activate *Drosophila* Imd innate immune response via upregulating the expression of the NF-κB transcription factor Relish [9, 10]. The E3-ligase inhibitor of apoptosis 2 (Iap2) could activate Dredd expression to positively regulate the Imd immune response [11, 12]. Furthermore, Imd-mediated immune response can be negatively regulated by some immune suppressors, such as WntD, Die, PGRP-LF, pirk, dUSP36, CYLD, Dnr1, dRYBP and Caspar [13–23], which can prevent the excessive activation of *Drosophila* Imd pathway to maintain innate immune homeostasis. In addition, many miRNAs have been reported to participate in fine-tuning *Drosophila* Imd immune response positively or negatively. Studies have shown that *Drosophila* miR-8 and miRNA let-7 could negatively regulate the Imd pathway [24, 25], whereas miR-34 could positively regulate the Imd pathway [8]. Our previous works have also demonstrated that both miR-9a and miR-981 could negatively regulate *Drosophila* Imd-dependent immune response via directly targeting the *AMP* gene *Diptericin (Dpt)* [26]. Although several regulators involved in *Drosophila* innate immune responses have been identified, the further study of the restoration mechanism of *Drosophila* innate immune homeostasis is still needed.

Myc serves as a broad-spectrum transcription factor to control the expression of a large number of genes for diverse cellular processes, including cell cycle progression, cell growth, proliferation and apoptosis [27–31]. Myc family includes three member of c-Myc, N-Myc, and L-Myc in human [27]. It's well documented that *Myc* can function as a proto-oncogene in human cancers [32, 33]. *Drosophila* has only one single *Myc* gene, referred to as *dMyc* or *diminutive* (*dm*), which is homologous to human *c-Myc* [34]. Studies have revealed that dMyc could involve in ribosome biogenesis [35], protein synthesis [36, 37], cell-autonomous apoptosis [38, 39], and cell competition [40, 41]. Although Myc can regulate the expression of some miRNAs to participate in immune response in human [42–48], it's largely unknown whether and how dMyc can regulate miRNA expression to control innate immune response in *Drosophila*.

In this work, we firstly report that dMyc can inhibit the immune response of *Drosophila* Imd pathway by genetic screening. Secondly, we confirm that dMyc directly activate the transcription of *miR-277* using Chromatin immunoprecipitation (ChIP)-qPCR analysis and promoter reporter activity system. Thirdly, we find that miR-277 can negatively regulate *Drosophila* Imd signaling response via targeting and downregulating the expression of *imd* and *Tab2-Ra/b*, but not *Tab2-Rc*. We further verify that dMyc could negatively regulate *Drosophila* Imd signaling pathway via activating *miR-277* transcription to inhibit imd/Tab2 expression *in vivo* using dMyc and miR-277 SP co-highexpressed flies. Finally, we provide evidences to show that dMyc could restore immune homeostasis of *Drosophila* Imd pathway to promote the survival of flies upon infection.

## Result

### dMyc is a negative regulator of *Drosophila* Imd pathway

*Drosophila* defends against Gram-negative bacteria infection through Imd pathway to produce *Diptericin* (*Dpt*) and other anti-microorganism peptides such as *Drosocin*, *Attacin* and *Cecropin A1*. In this work, via screening the fly strains from the Bloomington *Drosophila* library, we found *Gal80^ts-UAS-dmyc* flies significantly decrease the expressions of *Dpt*, *Drosocin*, *Attacin* and *Cecropin A1* after infection with gram-negative bacteria *Escherichia coli* (*E. coli)*, indicating dMyc regulates the Imd pathway (S1 Fig). Therefore, we here chose the *Dpt* as the representative for further exploring the regulatory role of dMyc in immune response of *Drosophila* Imd pathway. Our results demonstrate that the expression level of *Dpt* in the dMyc high-expressed flies with *E. coli* infection is, respectively, significantly lower than wild-type flies at all five time points (3, 6, 12, 24, 48h) (Fig 1A). The expression level of *dMyc* exists significant differences within different dmyc mutant fly strains (Fig 1B), and the order of the expression level of *dMyc* is dMyc high-expressed flies > dMyc and dMyc-RNAi co-highexpressed flies > wild-type flies > dMyc-RNAi high-expressed flies. Remarkably, the expression level of *Dpt* has no significant difference in these above dmyc mutant flies without infection (Fig 1C), but after infected with *E. coli*, the expression level of *Dpt* in dMyc-RNAi high-expressed flies is significantly higher than dMyc high-expressed flies and the control flies, respectively (Fig 1D). Especially, the expression level of *Dpt* in the dMyc and dMyc-RNAi co-highexpressed flies could be nearly restored to the normal expression level of the control flies (Fig 1D). Taken together, our findings strongly suggest that dMyc is a novel important negative regulator of *Drosophila* Imd pathway.

### dMyc regulates the expression of immune related miRNAs

To investigate whether and how dMyc negatively regulates Imd pathway via regulating miRNA expression, we used in silico analysis to identify potentially immune related dMyc-miRNA-gene axis (Fig 2A). We collected 63 genes involved in *Drosophila* Imd pathway from the literature data, and downloaded the maturation and upstream promoter sequences of *Drosophila* miRNAs from miRBase (http://www.mirbase.org/) and NCBI (https://www.ncbi.nlm.nih.gov/). The relationships between the 63 genes and miRNAs were next predicted by TargetScan and miRanda. Then miRNAs regulated by dMyc were predicted using the PROMO website and TransmiR 2.0 database.

We found 12 candidate dMyc-regulated miRNAs, which have been identified as differentially expressed miRNAs (DEmiRNAs) between flies with *E. coli* infection and the control from our previous small RNA-seq data [26] (Fig 2A, 2B and 2C). We next constructed these 12 DEmiRNAs high-expressed fly strains. We found that the expression level of *Dpt* is significant lower than the control after *E. coli* infection in miR-10, miR-1012, miR-277, miR-2b-2 and miR-996 high-expressed flies, respectively (Figs 2D and S2). We also investigated the expression relevance between dMyc and these 5 miRNAs, finding only miR-277's expression level in dMyc high-expressed flies is gone up 1.15 times than the control, but not miR-2b-2, miR-1012, miR-10 and miR-996 (Fig 2E). In contrast, the miR-277's expression level in dMyc knocked down flies is significant lower than the control (Fig 2F). In addition, two ChIP-seq data for dMyc from ENCODE database (https://www.encodeproject.org/) indicated that dMyc could bind to the upstream promoter sequences of *Drosophila miR-277* gene (S3 Fig). Taken together, our results suggest that dMyc might regulate miR-277 expression to negatively control Imd signaling pathway.

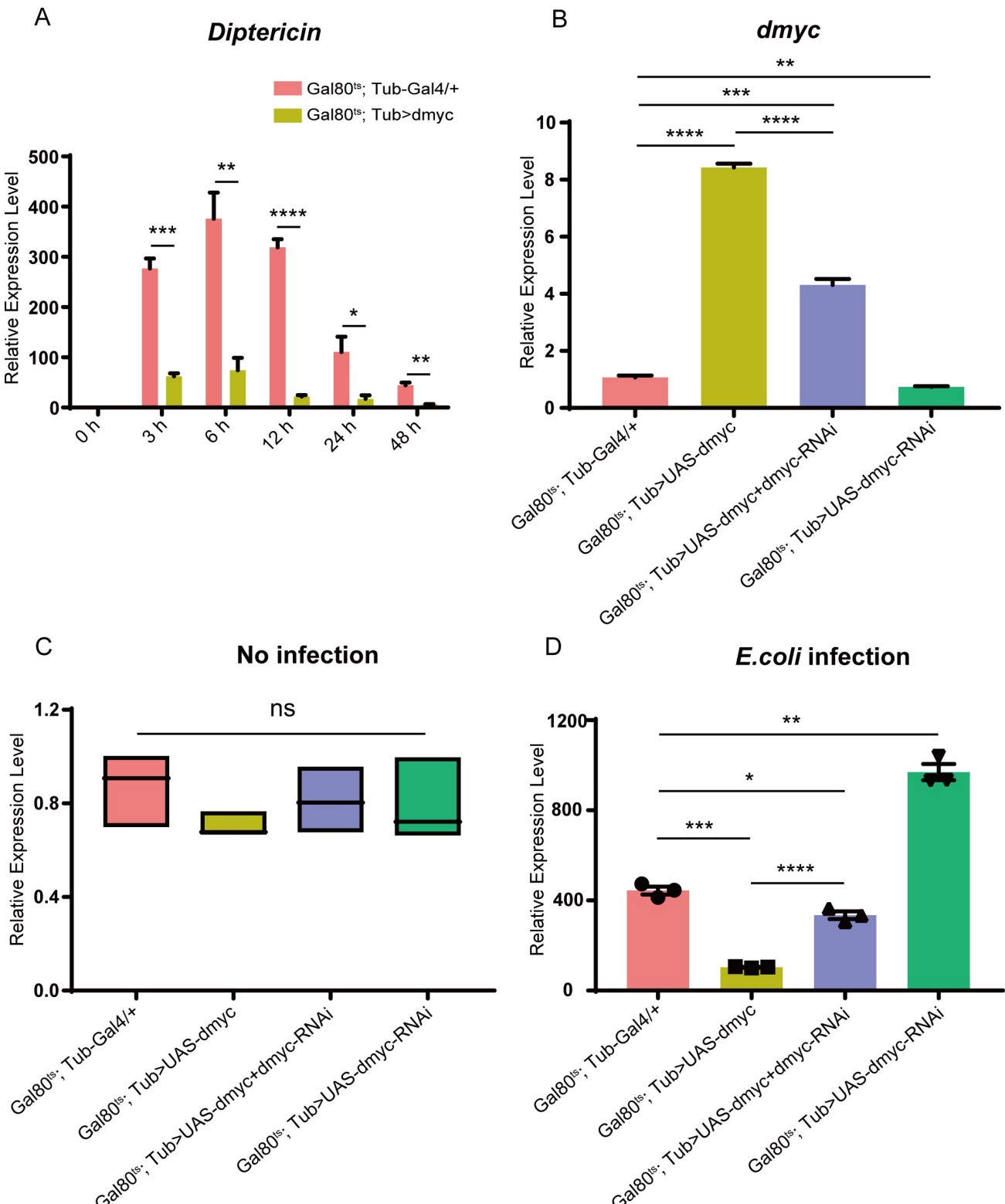

**Fig 1. The dMyc functions as a negative regulator of *Drosophila* Imd pathway.** (A) The transcript level of *Dpt* in dMyc highexpressed flies was measured at 0, 3, 6, 12, 24 and 48 h upon *E. coli* infection. (B) The expression level of *dMyc* was examined in control flies (*Gal80^{ts}; Tub-Gal4/+*), dMyc high-expressed flies (*Gal80^{ts}; Tub>UAS-dmyc*), dMyc and dMyc-RNAi co-highexpressed flies (*Gal80^{ts}; Tub>UAS-dmyc+dmyc-RNAi*) and dMyc-RNAi alone high-expressing flies (*Gal80^{ts}; Tub>UAS-dmyc-RNAi*) before *E. coli* infection. (C) The expression level of *Dpt* was detected in control flies (*Gal80^{ts};*

*Tub-Gal4/+*), dMyc high-expressed flies (*Gal80^{ts}; Tub>UAS-dmyc*), dMyc and dMyc-RNAi co-highexpressed flies (*Gal80^{ts}; Tub>UAS-dmyc+dmyc-RNAi*) and dMyc-RNAi alone high-expressing flies (*Gal80^{ts}; Tub>UAS-dmyc-RNAi*) before *E. coli* infection. (D) The expression levels of *Dpt* both in the dMyc and dMyc-RNAi co-highexpressed fly strains (*Gal80^{ts}; Tub>UAS-dmyc+dmyc-RNAi*) and the dMyc-RNAi high-expressing fly strains (*Gal80^{ts}; Tub>UAS-dmyc-RNAi*) were determined at 6 h upon *E. coli* infection.

## dMyc directly activates the transcription of *miR-277*

To further study how dMyc regulates miR-277, we performed an analysis for the upstream promoter sequences of *miR-277* and found that *miR-277* gene contains 2 transcriptional start sites (TSSs) [49] (Fig 3A). The upstream sequences of these two TSSs have the promoter activity, and the promoter activity of TSS1 upstream sequence is stronger than the TSS2 (Fig 3B). Lipopolysaccharide (LPS) stimulation could enhance these two promoter activities to increase *miR-277* expression (Fig 3B). Furthermore, dMyc could enhance the luciferase activities of these two promoter regions, and the promoter activity of TSS1 upstream sequence is consistently stronger than that of TSS2 (Fig 3C). ChIP-qPCR assay showed that dMyc is enriched at three candidate regions in the promoter sequences, and the neighbor region of TSS1 (ChIP1) is most enriched (Fig 3D). Taken together, our results suggest that dMyc activates *miR-277* transcription via directly binding to its promoter.

## miR-277 inhibits the expression of *imd* and *Tab2-Ra/b*

To further determine how miR-277 regulates the Imd pathway, first, we examined the expression level of *Dpt* in miR-277 high-expressed flies compared to the control group flies upon infection. The result showed that the expression level of *Dpt* in miR-277 high-expressed flies is significantly down-regulated at 3, 6 and 12 h post-infection compared with the control (Fig 4A). Moreover, the miR-277 rescue assay showed that this miR-277 and miR-277 sponge co-highexpressed flies could restore the expression level of miR-277 to the normal level (Fig 4B). Without infection, the expression level of *Dpt* has no significant difference between miR-277 mutant flies and the control group flies (Fig 4C). Remarkably, the expression level of *Dpt* could be recovered to the comparable level of the control group in miR-277 and miR-277 sponge co-highexpressed flies after *E. coli* infection (Fig 4D). Taken together, these results confirm that miR-277 negatively regulates the Imd pathway.

To further explore how miR-277 inhibits the Imd pathway, we predicted the potential target genes of miR-277 using targetScan and miRanda. We found that miR-277 could target to the 3'UTR of *imd* and *Tab2 Ra/b* transcripts (Fig 5A and 5B). *Imd* and *Tab2 Ra/b* are key components of the Imd pathway in *Drosophila* [6, 50]. As expected, the expression levels of *Dpt* in both imd-RNAi and Tab2-RNAi mutant flies are significantly lower than the control upon *E. coli* infection (S4A and S4B Fig). Consistently, compared with the control, the expression levels of both *imd* and *Tab2* are also significantly down-regulated in flies with high-expressed miR-277 at 3 h, 6 h and 12 h post infection, respectively (Fig 5C and 5D). In addition, the expression levels of *imd* and *Tab2* in miR-277 and miR-277 sponge co-highexpressed flies could be restored to the level of the control group (Fig 5E and 5F). Our findings suggest that miR-277 could inhibit the expression of both *imd* and *Tab2 in vivo*.

To further evaluate the direct targeting relationship between miR-277 and *imd* as well as *Tab2*, we carried out the Luciferase Reporter Assay in *Drosophila* S2 Cell. The results showed that compared with the pAc empty vector, miR-277 could significantly reduce the activity of the luciferase reporter containing the 3'UTR of *imd* and *Tab2 Ra/b*, but not *Tab2 Rc* (Fig 5G and 5H). Furthermore, we performed the target site mutation in the 3'UTR of *imd* and *Tab2 Ra/b*, finding that the reporter activity of *imd* and *Tab2 Ra/b* could be nearly restored to the normal level in these cells with co-transfected miR-277 expression vector and 3'UTR mutant

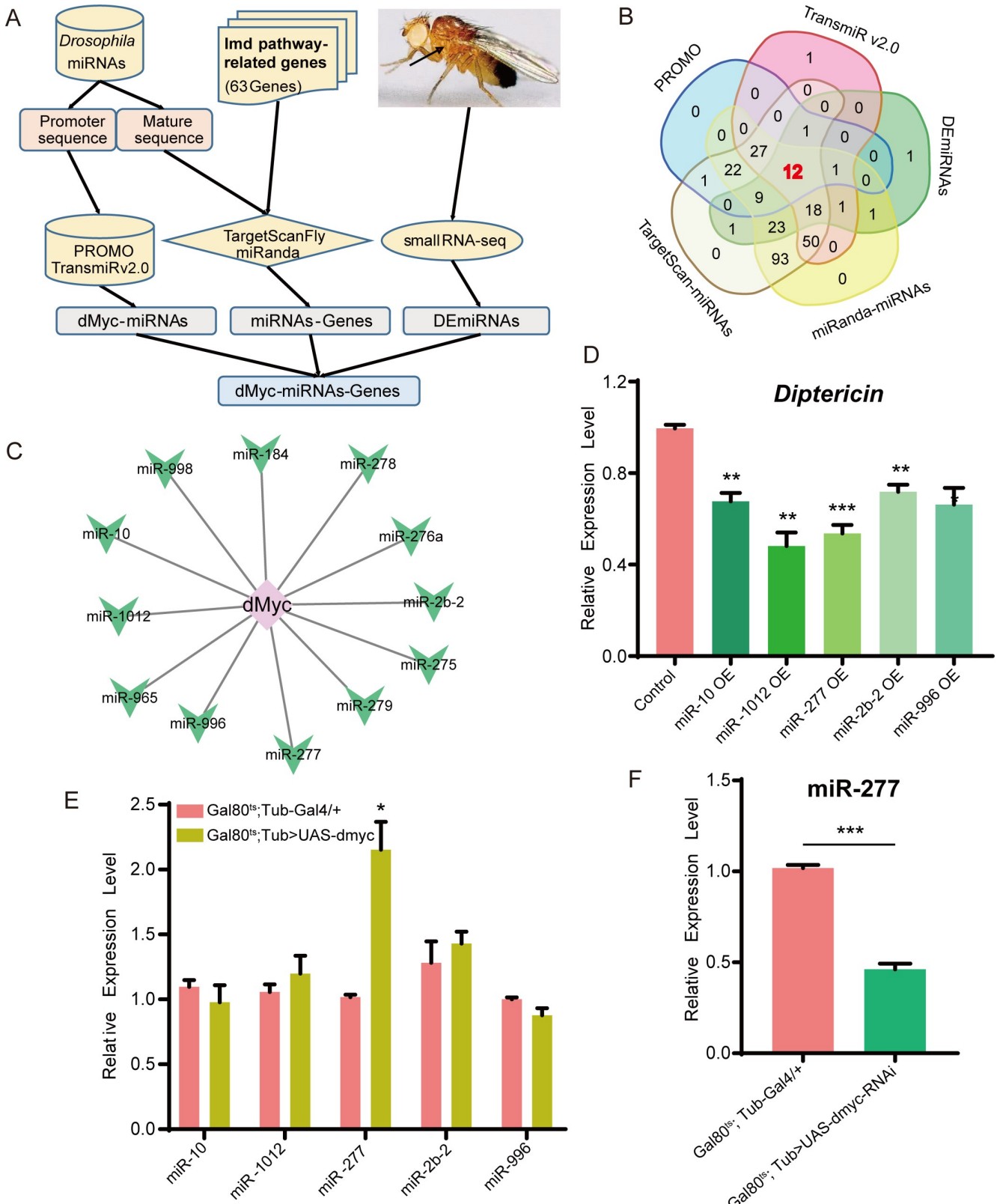

**Fig 2. The screening of dMyc regulating *Drosophila* immune related miRNAs.** (A) The flowchart showed the process of obtaining a batch of potential immune-related dMyc-miRNAs-Genes axes using bioinformatics tools. (B) The Venn diagram exhibited that the intersection of miRNAs that dMyc

regulating *Drosophila* immune-related miRNAs. (C) These 12 miRNAs regulated by dMyc were presented as a network diagram. (D) After these above miRNAs were high-expressed respectively, the expression level of *Dpt* was determined at 6 h upon *E. coli* infection. OE: overexpression. (E) The expression level of corresponding miRNA was detected in dMyc highexpressed flies. (F) The expression level of miR-277 was measured in dMyc-RNAi highexpresed flies.

reporters of *imd* and *Tab2 Ra/b* (Fig 5G and 5H). Taken together, our results suggest that miR-277 directly targets the 3'UTR of *imd* and *Tab2 Ra/b*.

## dMyc negatively regulates *Drosophila* Imd pathway via activating miR-277 to inhibit *imd*/*Tab2*

To further ascertain whether dMyc can regulate the *Drosophila* immune response via activating the transcription of *miR-277 in vivo*, we constructed the dMyc and miR-277 SP co-highexpressed mutation flies. We found that the expression level of miR-277 in the dMyc and miR-277 SP co-highexpressed flies is significantly lower than the dMyc highexpressed flies, and is nearly restored to the control level (S5A and S5B Fig). Without infection, the expression level of *Dpt* shows no significant difference in aforesaid mutant flies, and the expression level of *Dpt* in this dMyc and miR-277 SP co-highexpressed flies is significantly higher than the dMyc highexpressed flies upon infection (Fig 6A and 6B). In addition, the expression level of *Dpt* in the dMyc and miR-277 SP co-highexpressed flies at 6 h post infection could be restored to 55% of the control level, and is nearly 3 times than the dMyc highexpressed flies (Fig 6B). The expression levels of *imd* and *Tab2*, respectively, are also restored to 83% and 58% of the control level, and are 2.1 and 1.8 fold than the dMyc highexpressed flies, respectively (Fig 6C and 6D). Taken together, our data suggest that dMyc negatively regulates *Drosophila* Imd immune response via activating *miR-277* transcription to inhibit *imd*/*Tab2* expression.

## dMyc controls *Drosophila* Imd immune homeostasis

dMyc could negatively regulate *Drosophila* Imd immune response, indicating dMyc could be involved in the maintenance of Imd immune homeostasis. To test this point, we further monitored the dynamic expressions of *Dpt*, *dMyc*, miR-277, *imd* and *Tab2* in this wild-type flies at 0, 3, 6, 12, 24 and 48 h after *E. coli* infection. Our results showed that the expression level of *Dpt* is increasing before 3 h and reaching its peak expression level at 12 h, then gradually decreased to the basal level post-infection (Fig 7A). In contrast, the expression level of *dMyc* is decreased before 6 h, implying that the expression of *dMyc* is inhibited for avoiding the inadequate of immune response in the early stage of *E. coli* infection (Fig 7B). The expression level of *dMyc* is markedly up-regulated at 12 h post-infection, and subsequently is restored to the pre-infection level (Fig 7B). Moreover, the expression pattern of miR-277 is very similar with that of *dMyc*, whereas the expression patterns of both *imd* and *Tab2* are opposites of that of dMyc and miR-277 (Fig 7C and 7D). Taken together, we propose that dMyc could play a key role in restoring *Drosophila* Imd immune homeostasis post infection.

Immune homeostasis post infection is essential to organisms. We hypothesized that dMyc could protect *Drosophila* from damage caused by over-activation of immune response. To test this, we further investigated the survival rate of dMyc high-expressed flies and the control (Gal80$^{ts}$; Tub-Gal4/+) flies without infection and with PBS as well as the Gram-negative bacteria *Enterobacter cloacae* (*E. cloacae*) infection, respectively (Fig 8). We found that the survival rate of dMyc high-expressed flies is significantly lower than the control group both in the absence of infection and after infection with PBS. However, the survival rate of dMyc high-expressed flies is significantly higher than the control group after infection with *E. cloacae*.

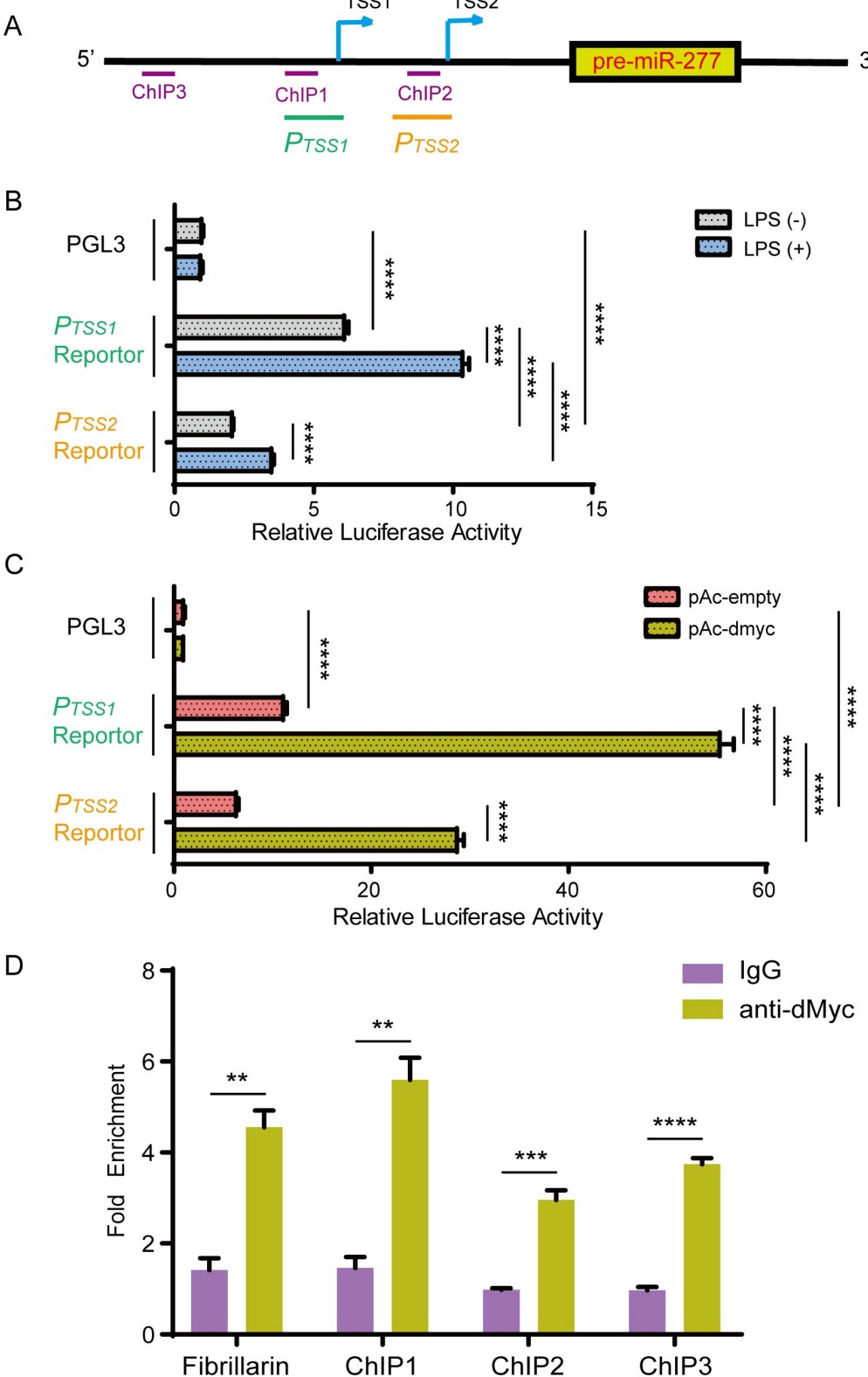

**Fig 3. dMyc activates the transcription of *miR-277*.** (A) According to the two TSSs of the *miR-277* promoter region, the $P_{TSS1}$ (green line) and $P_{TSS2}$ (orange line) region were selected to perform promoter activity analysis, while the *ChIP1*, *ChIP2* and *ChIP3* sequences (purple line) were selected to design the primers for ChIP-qPCR assay,

respectively. (B) The $P_{TSS1}$ and $P_{TSS2}$ reporter activity were determined with or without LPS stimulation, in *Drosophila* S2 cell on a luciferase assay. (C) The $P_{TSS1}$ and $P_{TSS2}$ reporter activity were determined, together with or without exogenous dMyc, in *Drosophila* S2 cell on a luciferase assay. (D) The fold enrichment of *ChIP1*, *ChIP2*, and *ChIP3* region were measured by ChIP-qPCR, respectively.

These results seem to support the important role of dMyc in negatively regulating the Imd pathway for immune homeostasis, which is essential for fly survival.

Taken all results together, we proposed a molecular mechanism by which dMyc plays an important role in *Drosophila* Imd immune homeostasis (Fig 9). On the one hand, down-expressed dMyc could down-regulate miR-277 expression to ensure the elevated expression of *imd* and *Tab2* at the early stage of *E. coli* infection to promote the expression of *Dpt* against pathogenic bacteria. On the other hand, to prevent the overactivation of Imd immune response, over-expressed AMP *Dpt* induces dMyc expression to activate the expression of miR-277 for down-regulating the expression of *imd* and *Tab2* to reduce *Dpt* expression, which restores Imd immune response to homeostasis to protect *Drosophila* from damage caused by overactivation of immune response, and improve the survival of *Drosophila*.

## Discussion

The *Drosophila* innate immune system plays critical roles in defending invading pathogens. Depression and overactivation of innate immune responses are both harmful for *Drosophila*. Thus, the *Drosophila* innate immune system must gain an unknown mechanism to resist path-ogen challenges without overactivation of innate immunity. Although studies have revealed that the *Drosophila* innate immune response could be controlled by a series of negative or posi-tive regulators at transcriptional and post-transcriptional levels [8, 24–26, 51, 52], the mecha-nism of maintaining immune homeostasis is largely unknown. In this study, we reveal a new dMyc-miR-277-imd/Tab2 axis to play an important role in negatively regulating Imd pathway, and provide a mechanistic insight into immune homeostasis in *Drosophila*.

We found that high-expressed dMyc led to decrease of *Dpt* expression, conversely knock-down dMyc resulted in increase of *Dpt* expression (Fig 1B), indicating that dMyc act as a nega-tive regulator of *Drosophila* Imd pathway. Previous studies have revealed that human Mycs play key roles in activating both innate and adaptive immune cells to defense invading patho-gens [1, 53, 54]. Functions of Myc are evolutionarily conserved between fruit fly and vertebrate [55]. The fruit flies and vertebrate proteins can substitute for each other to the extent [56]. Thus, our study provides an important insight into illuminate the conservative immune regu-lation function of Myc between *Drosophila* and human.

Human Mycs, as important transcriptional factors, not only can control the expressions of a large number of protein-coding genes, but can regulate the expressions of many miRNAs [57, 58]. In this work, we further identified *miR-277* as a target gene of dMyc (Fig 3), and found miR-277 colud negatively regulate *Dpt* expression by targeting *imd* and *Tab2* in *Dro-sophila* Imd immune responses (Fig 6). Previous studies have indicated that *imd* and *Tab2* are specifically required for the immune activation of *Drosophila* Imd signaling pathway [6, 59]. Thus, our results suggest that dMyc might negatively regulate *Drosophila* Imd immune response via activating *miR-277* transcription to inhibit *imd*/*Tab2* expression. Previous reports have indicated that miR-277 not only can control branched-chain amino acid catabolism, affect lifespan [60] and wing imaginal discs development of *Drosophila* [61], but modulate the Neurodegeneration Caused by Fragile X Premutation rCGG Repeats [62]. Whereas, our pres-ent results demonstrate that miR-277 is a new negative regulator involved in *Drosophila* Imd signaling pathway. Especially, the *Tab2* is an alternative splicing gene, which contains three

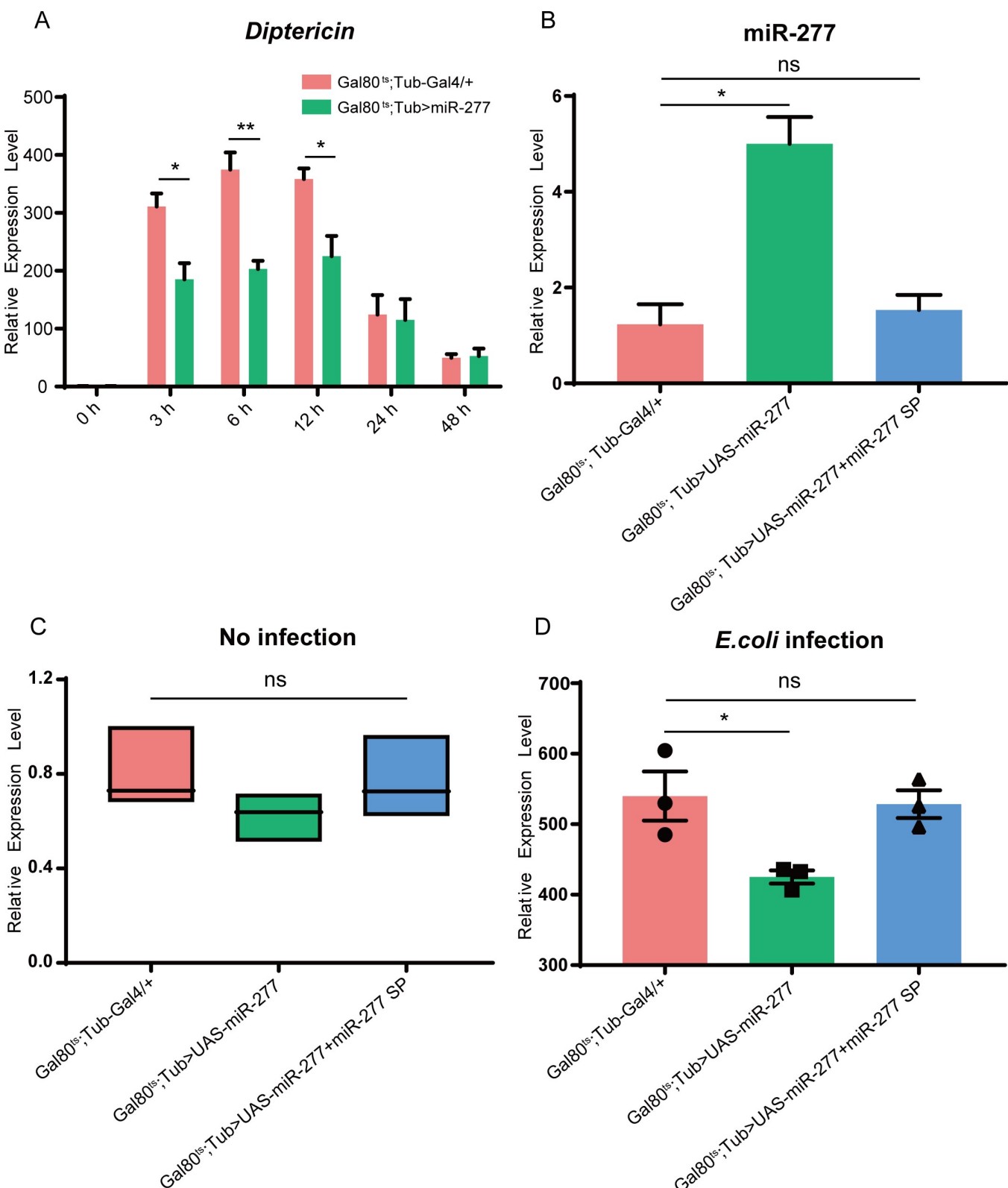

**Fig 4. miR-277 negatively regulates *Drosophila* Imd pathway immune response.** (A) The expression level of *Dpt* was detected in miR-277 highexpressed fly strains (*Gal80^ts^; Tub>miR-277*) at 0, 3, 6, 12, 24 and 48 h upon *E. coli* infection. (B) The expression level of miR-277 was measured in the control flies, this miR-277 highrexpressed flies, and this miR-277 and miR-277 sponge co-highexpressed flies before *E. coli* infection. The transcriptional level of *Dpt* was determined before (C) and after (D) *E. coli* infection.

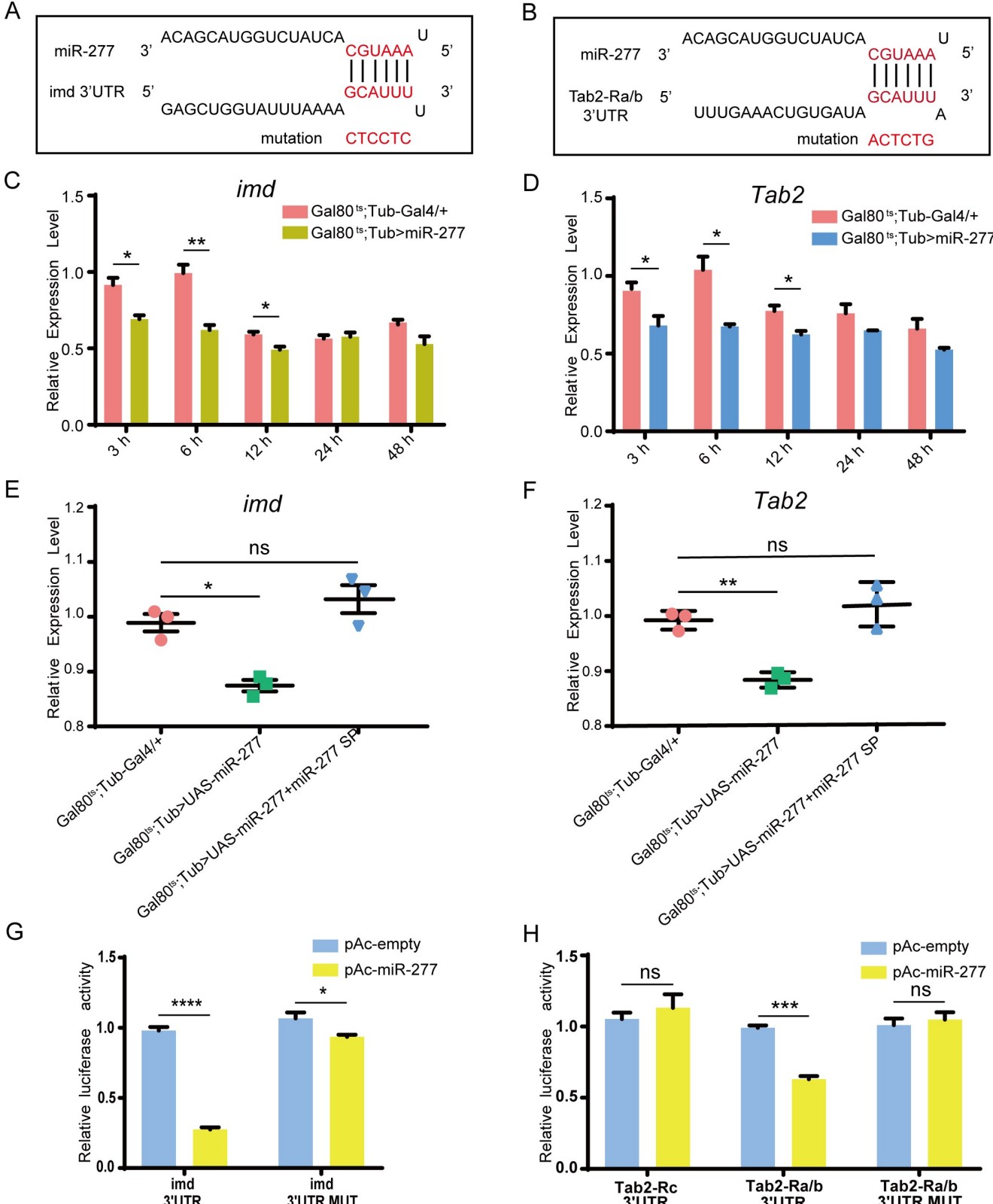

**Fig 5. miR-277 inhibits the expression of *imd* and *Tab2-Ra/b in vivo* and *in vitro*.** These potential binding sites of miR-277 in the 3'UTR of *imd* (A) and *Tab2-Ra/b* (B) were predicted by targetScan and miRanda software. These point mutations at the 3'UTR target sites base pairing to the seed sequence of

miR-277 were performed. These expression level of *imd* (CE) and *Tab2* (DF) were respectively tested in miR-277 high-expressing flies and miR-277 and miR-277 sponge co-highexpressed flies. The corresponding luciferase activity of the report plasmids without or with mutation sites was determined in *Drosophila* S2 cell on a Dual luciferase assay (GH).

transcripts, i.e. *Tab2 Ra*, *Tab2 Rb* and *Tab2 Rc*, of which *Tab2 Ra* and *Tab2 Rb*'s 3'UTR is identical. Here, our results showed that miR-277 can target to the 3'UTR of *Tab2 Ra* and *Tab2 Rb* transcripts, but not *Tab2 Rc* (Fig 5). This result suggests that miRNA might play a critical role in selectively regulating the expression of alternative splicing gene in the post-transcriptional level.

Innate immune is a rapid and short immune response process, and inactivation or overactivation of innate immune responses can result in the normal tissue damage [63–66]. We reported the tightly coordinated expression of *dmyc*, miR-277, *imd*,*Tab2 and Dpt*, suggesting that dMyc as a novel negative regulator primarily prevents the over-activation of *Drosophila* Imd immune response at this middle and later stage of *E. coli* infection, and helps *Drosophila* restore to a new immune homeostasis.

Recently, overexpression of dMyc has been reported to be able to significantly diminish *Drosophila* adult longevity, which might is due to over-expressed dMyc greatly resulting in genome instability [67]. In addition, studies have indicated that down-regulating expression level of c-Myc can significantly increase mice longevity due to heterozygosity [27, 68]. However, in our present study, we found that the overall survival rate of dMyc high-expressed adult male flies is similar with the control group at the early stage (0~5h) after *E. cloacae* infection (Fig 8). Whereas, the survival rate of dMyc high-expressed adult male flies was significantly higher than the control group after 5 h post infection (Fig 8). Taken together, our findings suggest that dMyc contributes to the survival of flies likely via preventing over-activation of innate immune responses to avoid excessive damage of many tissues.

## Conclusions

In this work, we identify dMyc as a novel negative regulator of *Drosophila* Imd pathway. Mechanically, dMyc positively activate *miR-277* transcription, to target the 3'UTR of *imd* and *Tab2-Ra/b* to inhibit their expression, leading a new immune homeostasis. Our present results provide a new comprehensive understanding on the complex regulatory mechanism of maintaining innate immune homeostasis in *Drosophila*.

## Materials and methods

### Fly stocks

Flies were obtained from the Bloomington *Drosophila* Stock Center: *UAS-dmyc* (#7118); *UAS-miR-277* (#36559); *UAS-miR-277^sponge^* (#61408). These Fly lines carrying UAS-RNA interference (RNAi) constructs were obtained from the Tsinghua Fly Center: *dmyc^RNAi^* (#47953); *imd^RNAi^* (#31706); *Tab2^RNAi^* (#24667). As well as previously purchased tubulin-Gal80^ts^;TM2/TM6B (#7019) and tubulin-Gal4/TM3, Sb[1], Ser[1] (#5138), all flies were raised on maize malt molasses food in a light-dark (12-hr cycle) incubator at 25°C and 60% humidity. Flies were shifted to 30°C 24 h prior to and then during infection for *UAS-protein*/*UAS-miR-277^sponge^*/*UAS-protein^RNAi^* overexpression experiments.

### Bioinformatic analysis

These mature sequences of *Drosophila* miRNAs were downloaded from miRBase ([http://www.mirbase.org/](http://www.mirbase.org/)). 3'UTRs of 63 Imd pathway-related genes and promoter sequences of *Drosophila*

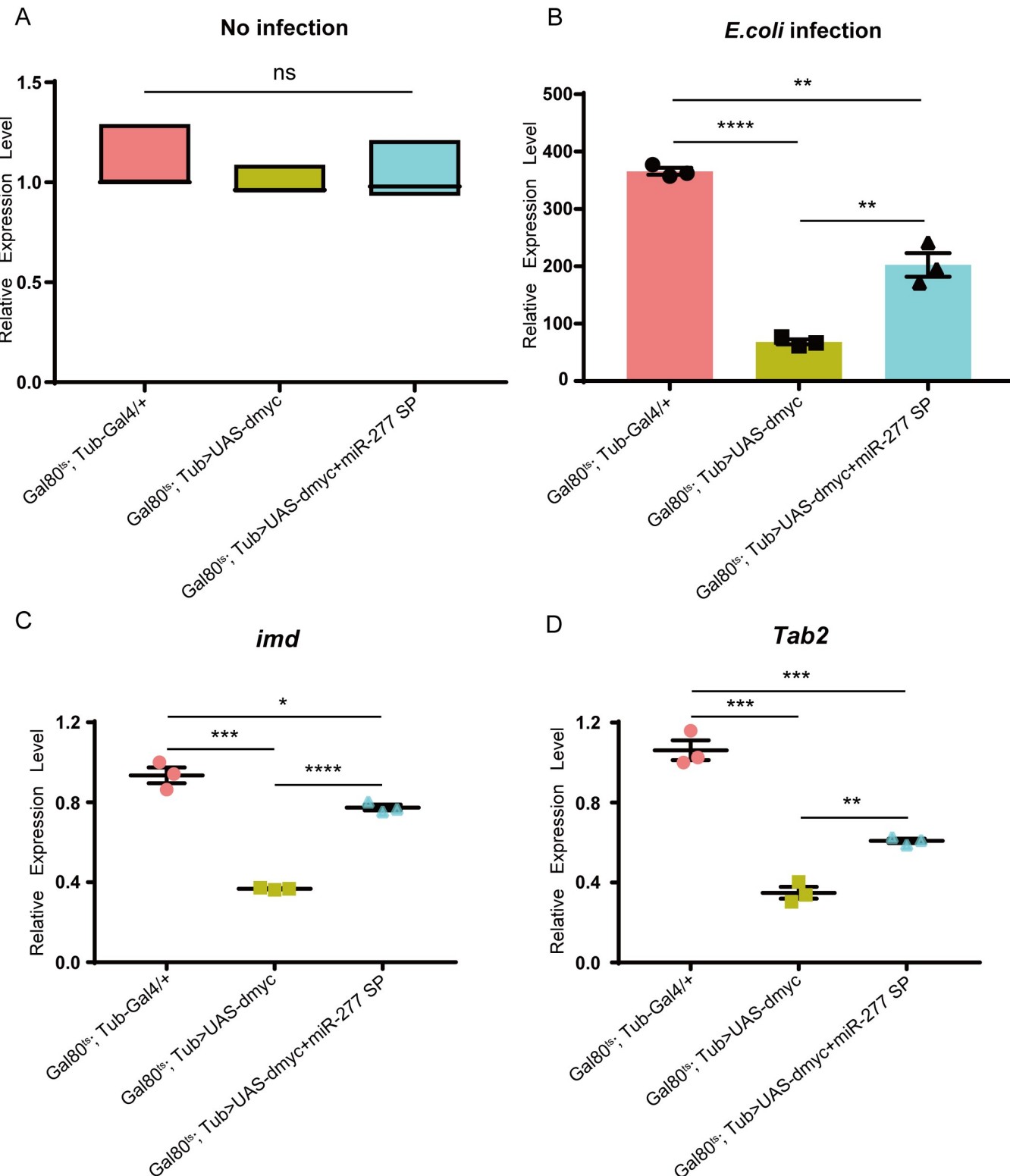

**Fig 6. dMyc negatively regulates *Drosophila* Imd pathway via activating miR-277 to inhibit *imd*/*Tab2* expression.** (A) The expression level of *Dpt* was examined in control flies, dMyc high-expressed flies, dMyc and miR-277 sponge co-highexpressed flies (*Gal80ts*; *Tub>UAS-dmyc+miR-277 SP*) before *E. coli* infection, respectively. The expression levels of *Dpt* (B), *imd* (C) and *Tab2* (D) were detected and compared in dMyc highexpressed flies and dMyc and miR-277 co-highexpressed flies at 6 h following *E. coli* infection, respectively.

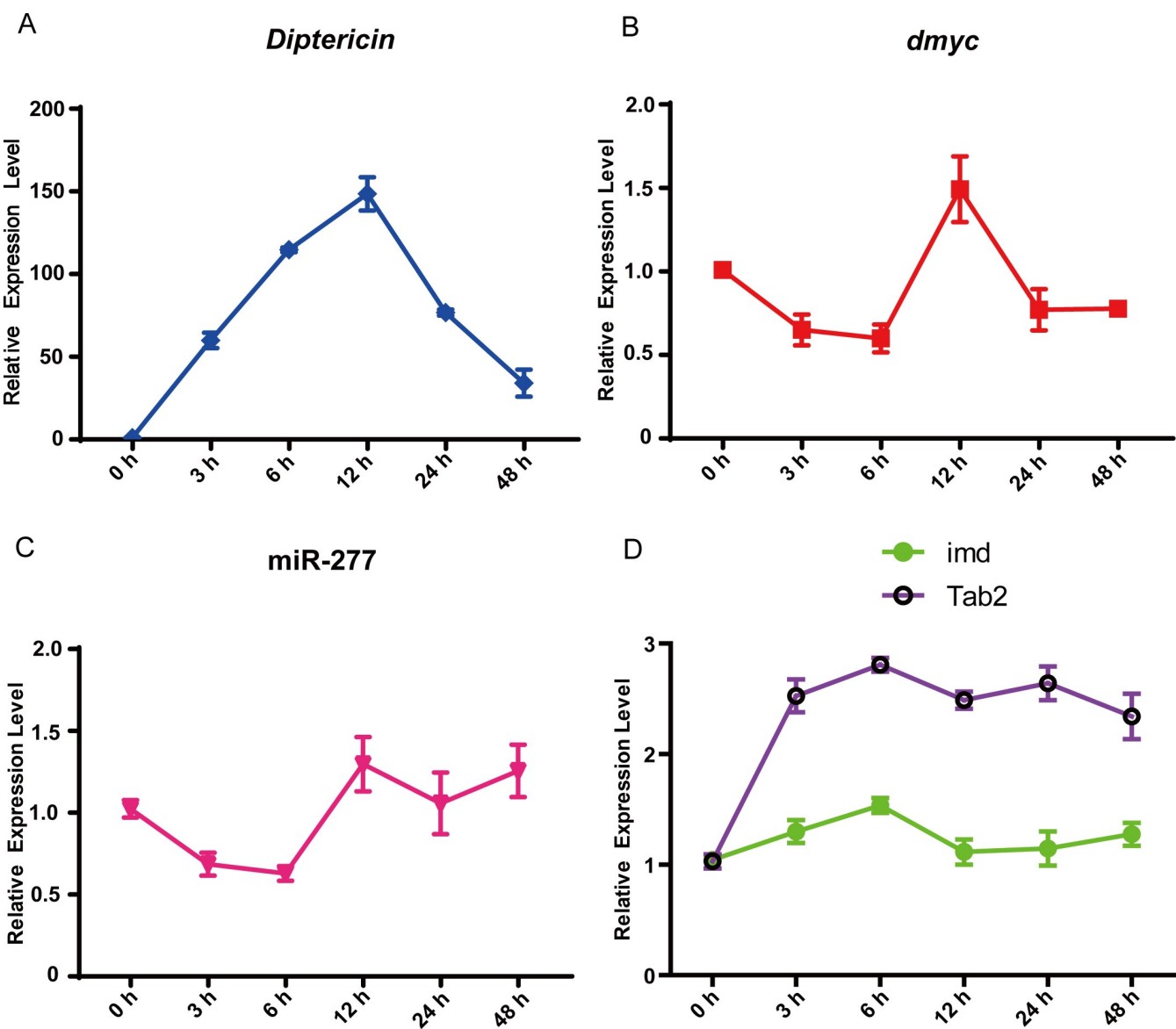

**Fig 7. The temporal expression patterns of four genes and miR-277 in the wild-type flies prior to and following *E. coli* infection.** The dynamic expression changes of *Dpt* (A), *dmyc* (B), miR-277 (C), *imd* and *Tab2* (D) at six time-points (0, 3, 6, 12, 24 and 48 h) prior to and following *E. coli* infection, respectively.

miRNAs were extracted from the fruit fly genome in FlyBase (http://flybase.org/) and NCBI (https://www.ncbi.nlm.nih.gov/). These relationships between mature miRNAs and 3'UTR of genes were predicted using two miRNA target prediction programs with default parameters, i.e. TargetScan (www.targetscan.org/fly_12/) [69] and miRanda v3.3a tool downloaded from microRNA.org-Targets and Expression [70, 71]. Whilst these sites of transcription factor (dMyc) binding at these promoter sequences of miRNAs were predicted through PROMO website [72, 73] and TransmiR 2.0 database (http://www.cuilab.cn/transmir).

## Infection and survival experiments of adult flies

Three to four-day-old adult male flies were used for septic injury experiments. Control and high-expressed or knockdown gene/miRNA flies were infected by *E. coli*, which is a widely

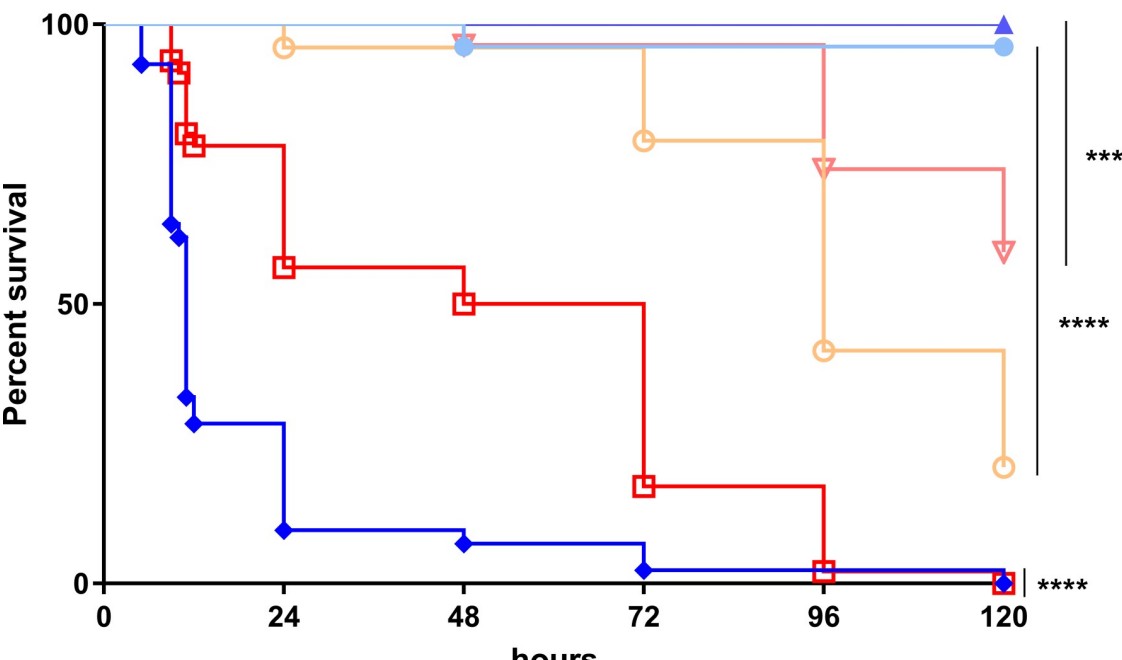

**Fig 8. The ectopic expression of dMyc influences the survival of *Drosophila*.** The changes of the survival rate were observed in dMyc highexpressed flies and the control (Gal80$^{ts}$;Tub-Gal4/+) flies without infection and with PBS as well as *E. cloacae* infection; dMyc OE (overexpression): Gal80$^{ts}$;Tub>UAS-dMyc.

used bacterial strain that can activate the Imd-mediated immune response to induce the expression of *Diptericin*. Infection experiments were performed by pricking the thorax of the flies with a pulled glass capillary carrying *E. coli* inoculant using a Nanoject apparatus (Nanoliter 2010, WPI). Next, flies were collected at specified time-points for subsequent experiments. Survival to infection is the most holistic approach to assess these defects in immune response [74]. For the survival experiment, flies were infected with a concentrated culture of *E. cloacae* by pricking as above, and then the survival situation of flies was detailedly recorded for 5 days.

## RNA extraction and RT-qPCR

Total RNAs were isolated from these treated adult flies using TRIzol Reagent (Invitrogen) following the instructions. For RT-PCR, A first-strand cDNA synthesis kit (Vazyme, China) was used to prepare the cDNA. These stem-loop primers were synthesized for reverse transcription to generate the specific stem-loop cDNA of miRNA. Quantitative PCR reactions were performed using AceQ SYBR Green Master Mix (Vazyme, China) on the ABI StepOne Plus Real-

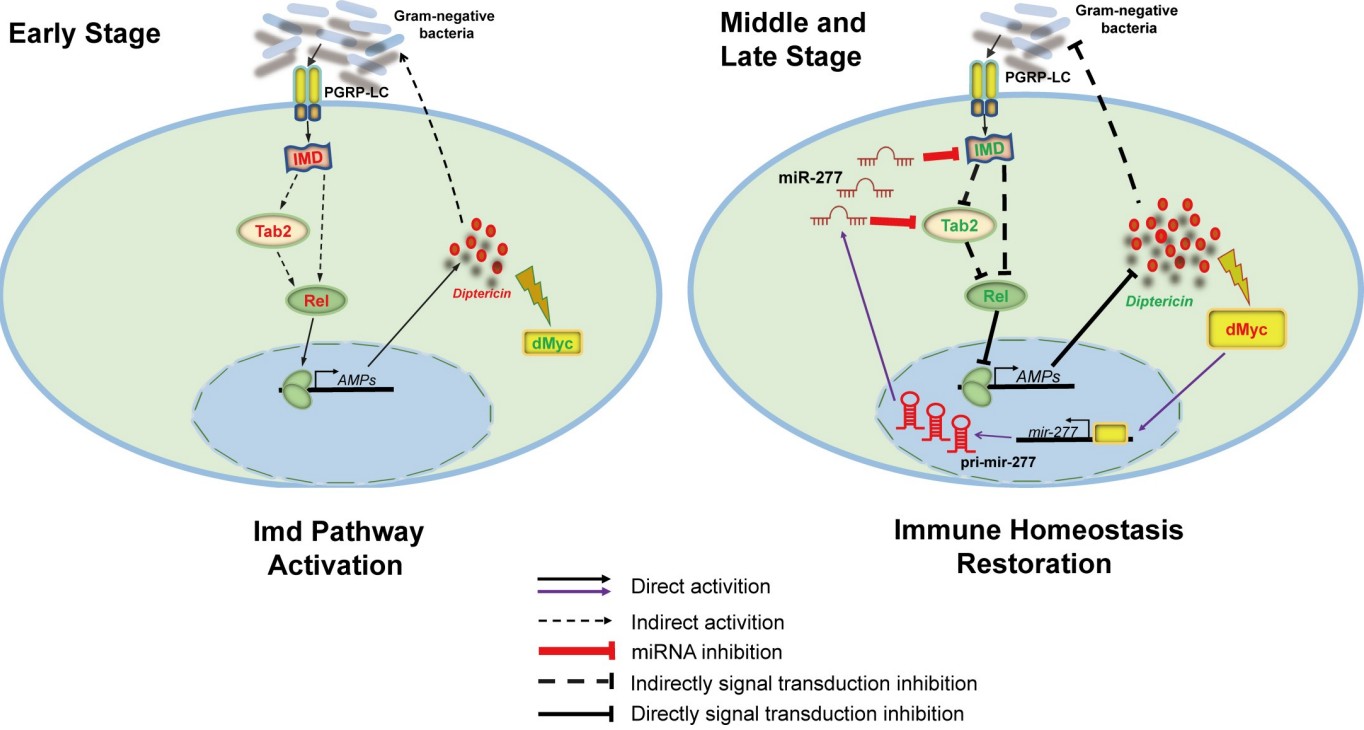

**Fig 9. A potential molecular mechanism of dMyc restoring immune homeostasis of *Drosophila* Imd pathway.** Our results suggested a model in which dMyc could restore *Drosophila* Imd immune homeostasis at the middle and late stage of *E. coli* infection. Left diagram: *Drosophila* Imd pathway is activated at the early stage of infection. Right diagram: Most of the bacteria have been eliminated at the middle stage of infection, next highly expressed antimicrobial peptides induce the up-regulated expression of dMyc. Then dMyc transfers into the nuclear to bind the upstream region of *miR-277* to activate *miR-277* transcription. Mature miR-277 further suppresses *imd* and *Tab2* expression to down-regulate the Imd pathway immune response, and then assists cells restore immune homeostasis at the late stage of infection. Red word: up-regulation; Green word: down-regulation.

Time PCR System (Applied Biosystems, USA). The expression levels of mRNA and miRNA were normalized to the control rp49 and U6 snRNA, respectively. All experiments were in triplicate. The relative $2^{-\triangle\triangle CT}$ method was used for data analysis [75]. All primers used in this analysis were listed in S1 Table.

## Cell culture and immune stimulations

*Drosophila* S2 cells were maintained at 28˚C in Schneider's medium (Invitrogen) supplemented with 10% fetal bovine serum (FBS) and 1% penicillin-streptomycin (Invitrogen). For immune stimulation, cells were incubated with 10μg/ml commercial LPS from *E. coli* 055:B5 (Sigma, St. Louis, MO), which is characteristic components of the cell wall of Gram-negative bacteria, for 6 h [76, 77].

## Chromatin immunoprecipitation (ChIP)

For ChIP experiment, Cells were fixed by cross-linking with a final concentration of 1% formaldehyde solution for 10 min at room temperature and then quenched with 125 mM glycine for 5 min. After washing with cold PBS containing a protease inhibitor cocktail and PMSF twice, these cells were lysed with cell lysis buffer and nuclear lysis buffer. The clarified lysate was subject to sonication. The chromatin was then sheared to fragments of 200–500 bp. The chromatin was used for ChIP incubating with Dynabeads protein G (Thermo Fisher Scientific) coated with either an anti-dMyc antibody (P4C4-B10; DSHB) or mouse IgG control antibody

overnight at 4˚C on a rotating platform. After repeated washes using a magnetic rack (Thermo Fisher Scientific), dMyc-bound genomic DNA was eluted from Dynabeads, and then the cross-links were reversed at 65˚C for 4h (or overnight). DNA fragments then were purified with AxyPrep PCR Cleanup Kit (Axygen). qRT-PCR analysis was performed using the DNA from the Input and ChIP experiments with primers listed in S2 Table. At least three independent experiments were carried out for the miR-277 promoters, as well as for *Fibrillarin* gene served as a positive control [78].

## Luciferase reporter construction and luciferase assay

This *pri-mir-277* has two transcription initiation sites (TSSs) as reported [49], so we further divided the promoter region of *pri-mir-277* into two parts for luciferase promoter analysis. Promoter sequences of *miR-277* and CDS of *dMyc* were amplified by PCR from *Drosophila* genomic DNA. The DNA fragments were then isolated and inserted respectively into the restriction enzyme digested the promoterless pGL3 Basic and pAc5.1 Vector using T4 DNA ligase. pAc5.1 luciferase reporter constructs carrying the 3'-UTR of either *imd* or *Tab2* with wild-type or mutated sequences of their respective miR-277 target sites were utilized to analyze the effects of the miR-277. All constructs were confirmed by sequencing. All PCR primers for the reporter constructs were listed in S3 Table. *Drosophila* S2 cells were transfected with each reporter construct for 48 h followed by assessment of luciferase activity. Luciferase activity was then measured with Dual Luciferase Reporter Assay System (Promega) according to the manufacturer's instructions and normalized to the Renilla luciferase activity for each transfected well. Each assay was performed in triplicate.

## Statistical analysis

All experimental data in this work were collected from three independent biological replicates. All statistical analyses were presented as means ± SEM. Significant differences between the values under different experimental conditions were subjected to two-tailed Student's t-test. Statistical analysis of fly survival experiments was performed using the log-rank (Mantel-Cox) test. For all tests, $P$ value $< 0.05$ was considered as statistically significant. $^*P < 0.05$; $^{**}P < 0.01$; $^{***}P < 0.001$; and ns, no significance vs. the control groups.

## Supporting information

**S1 Fig. The expression level of multiple *AMP*s in the dMyc high-expressing flies and the control flies.** The expression level of *Diptericin* (A), *Attacin* (B), *Cecropin A1* (C), and *Drosocin* (D) were measured in the dMyc high-expressing flies and the control flies upon *E. coli* infection.
(TIF)

**S2 Fig. The expression level of *Dpt* in 7 miRNA highexpressed fly strains.** After 7 miRNAs were high-expressed respectively, the *Dpt* level was determined at 6 h upon *E. coli* infection. OE: overexpression.
(TIF)

**S3 Fig. The bind sites of dMyc on the upstream of *miR-277* gene.** Two ChIP-seq data for dMyc from ENCODE database were visualized to show the bind sites of dMyc on the upstream of *miR-277* gene.
(TIF)

**S4 Fig. The expression level of *Dpt* in this imd-RNAi and Tab2-RNAi highexpressed fly strains.** The expression level of *Dpt* was determined respectively by RT-q PCR in this imd-RNAi (A) and the Tab2-RNAi (B) highexpressed flies upon *E. coli* infection.
(TIF)

**S5 Fig. The expression level of *dMyc* and miR-277 in dMyc and miR-277 sponge co-highexpressed fly strains.** The expression levels of *dMyc* (A) and miR-277 (B) were examined in the control flies, the dMyc highexpressed flies, the dMyc and miR-277 sponge co-highexpressed flies before *E. coli* infection.
(TIF)

**S1 Table. Primers used for quantitative RT-PCR**
(DOCX)

**S2 Table. Primers used for ChIP-qPCR**
(DOCX)

**S3 Table. Primers used for transgene vector construction**
(DOCX)

## Acknowledgments

We are grateful to the Bloomington Stock Center and the Tsinghua Fly Center for providing fly stocks.

## Author Contributions

**Conceptualization:** Ruimin Li, Ping Jin, Fei Ma.

**Data curation:** Ruimin Li, Fei Ma.

**Funding acquisition:** Fei Ma.

**Investigation:** Ruimin Li, Hongjian Zhou, Chaolong Jia.

**Methodology:** Ruimin Li, Hongjian Zhou, Fei Ma.

**Project administration:** Ping Jin, Fei Ma.

**Resources:** Ping Jin, Fei Ma.

**Supervision:** Ping Jin, Fei Ma.

**Validation:** Ruimin Li, Hongjian Zhou, Chaolong Jia.

**Visualization:** Ruimin Li, Ping Jin, Fei Ma.

**Writing – original draft:** Ruimin Li, Ping Jin, Fei Ma.

**Writing – review & editing:** Ruimin Li, Ping Jin, Fei Ma.

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
