## [Decision Letter · Decision Letter 0]

27 Jan 2020

Dear Dr Ma,

Thank you very much for submitting your Research Article entitled 'Drosophila Myc restores immune homeostasis of Imd pathway via activating miR-277 to inhibit imd/Tab2' to PLOS Genetics. Your manuscript was fully evaluated at the editorial level and by three independent peer reviewers. The reviewers appreciated the attention to an important problem, but raised some substantial concerns about the current manuscript. Based on the reviews, we will not be able to accept this version of the manuscript, but we would be willing to review a much-revised version. We cannot, of course, promise publication at that time.

The reviewers concerns were quite significant and a revised manuscript will need to include additional experimental work including, inclusion of proper controls, adding loss-of-function analysis to bolster claims that primarily rely on over-expression analysis, and substantial improvement of English grammar (this will likely require a professional editing service).

If you decide to revise the manuscript for further consideration at PLOS Genetics, please aim to resubmit within the next 60 days, unless it will take extra time to address the concerns of the reviewers, in which case we would appreciate an expected resubmission date by email to plosgenetics@plos.org.

[LINK]

We are sorry that we cannot be more positive about your manuscript at this stage. Please do not hesitate to contact us if you have any concerns or questions.

Yours sincerely,

Gregory P. Copenhaver

Editor-in-Chief

PLOS Genetics

Reviewer's Responses to Questions

**Comments to the Authors:**

Reviewer #1: In this manuscript, Li et al study the link between IMD pathway, Myc and miR-277

They propose that Myc represses the IMD pathway and that this is mediated by the miR-277.

There are many problems in this manuscript that are listed here.

- All the presented experiments analyzed the effects of Myc over expression using the Gal4 system. So, they do not study the role of Myc in the immune homeostasis but the consequences of Myc overexpression. This is therefore non-physiological and might have nothing to do with the endogenous role of Myc. Loss-of-function mutants should be use to study the role of a gene in a given process.

- Most of the experiments presented here use the diptericin as a read out of the IMD pathway. Hundreds of papers have shown that, as most Antimicrobial peptides, diptericin transcripts can increase us to 10.000 times upon infection. Even in the absence of infection, their levels vary quite a bit. The fold changes reported in this manuscript are around 1,5 or even sometimes less. There is something wrong here. I do not what. When a gene can have its transcript produced 10.000 times more in control conditions, what does a 1,3 fold difference mean? My guess is that there is a technical problem.

- The experiments always miss important controls. Just one example but this is true for all of them. Fig 8, the authors test the resistance of Myc overexpressing flies to E. Cloacae infection. The survival rate of these flies in the absence of infection and after sterile wounding should be tested.

- Immune genes can be activated by stress. Overexpressing the oncogene Myc in a cell is probably stressful for the flies. I am not sure this work studies immune response but rather stress response.

- The text is very difficult to follow. Just one example from the first sentence of the results “as we know that the diptericin expression is an important readout of Drosophila Imd pathway activation”. We expect the phrase to continue but it does not.

Reviewer #2: The manuscript “Drosophila Myc restores immune homeostasis of Imd pathway via activating miR-277 to inhibit imd/Tab2”(PGENETICS-D-19-02099) has revealed that Drosophila transcription factor dMyc could serve as a novel negative regulator of Imd pathway immune response using the loss- and gain-of-function screening. Especially, their works illuminate a feasible mechanism that dMyc as a transcription factor may activate miR-277 transcription to inhibit imd and/or Tab2 expression to restore immune homeostasis of Drosophila Imd pathway and improve the survival of flies. Overall, this manuscript has provided new insights into further studying the maintenance mechanism of innate immune homeostasis. The content of this manuscript is interesting. Thus I think that this manuscript is suitable to be published in PLOS Genetics. However, there are several shortcomings as following:

Major comments

Comment 1: In response to Gram-negative bacteria, the Imd pathway produces not only Dpt but also other antimicrobial peptides (AMPs). Thus, what are the expression levels of other AMPs in response to E.coli?

Comment 2: Although the expression level of dMyc in the dMyc and dMyc-RNAi co-highexpressed flies could be recovered to some extent, whether the expression level of miR-277 could also be restores?

Comment 3: Myc may recognize some binding motifs so-called E-boxes (e.g. CACGTG, CATGTG and alternative sequences) in the promoter region of target genes to activate their expression. Whether there is a similar E-box in the promoter region of miR-277? Especially, when the E-box of miR-277 is deleted or mutated, whether Myc could still activate the transcription of miR-277？

Comment 4: Why choose these five time points to collect and test samples after infection?

Minor comments

Comment 1: Some abbreviations, such as E.coli and LPS, should write the full name at the first appearance.

Comment 2: In Fig 3, the font of the title is inconsistent.

Comment 3: In Fig 7, line 326: “E.col” should be “E.coli”.

Comment 4: Line 430: “togther” should be “together”.

Comment 5: “(D)”should be “(B)” in Fig 1 legend (line 140).

Comment 6: “Drosophila” should be italic (line 116 and 148).

Comment 7: The legend in Fig4A is not consistent with other figures. It should be “Gal80ts” not “Gal80”.

Comment 8: Two legends in Fig 3B and 3C could look better when they are adjusted up and down.

Reviewer #3: Li et al. have studied the effect of Drosophila Myc on innate immune signaling. They have discovered that it acts as a negative regulator of the IMD pathway and they further show that it acts by inducing expression of miR-277 transcription which in turn down-regulates the expression of imd and Tab2-Ra/b (but not Tab2-Rc). Notably, dMyc was shown to bind to the promoter region of miR-277.

Major comments:

Most of all, the level of the English grammar – and the way of scientific writing – is not at the required level. As I see potential in this manuscript I hope that there would be careful editing for the paper for scientific English.

Figure 1 A, Dipt expression should be shown also in uninfected (0h) flies

Figure 3B: S2 cells respond poorly to LPS as in Drosophila, gram-negative bacteria are recognized by Peptidoglycan recognistion proteins, predominantly via PGRP-LC (use Choe et al., 2002 Nature, Ramet et al., 2002 Nature, Gottar et al., 2002 Science for references). I suggest repeating this experiments using heat-killed E. coli instead of LPS.

Minor comments:

Lines 75-77 ‘Furthermore, the Imd-mediated immune response can also be negatively regulated by some immune suppressors, such as WntD, Die, PGRP-LF, pirk, dUSP36, CYLD, Dnr1, dRYBP and Caspar [15-24]’ lacks the notion and reference to the most important negative regulator of this pathway, namely Pirk (references Kallio et al., 2005 Microbes and Infection and Kleino et al., J Immunol 2008 needs to be cited).

Line 117: ‘As we all know…’ Rephrase

Lines 124 Avoid phrases such as ‘remarkably’ – the results will speak to themselves

Lines 128-129: ‘Especially, the expression level of Diptericin in the dMyc and dMyc-RNAi co-highexpressed flies’ – I see no point – is I understand this correctly – to both overexpress dMyc and to suppress its expression by RNAi at the same time.

Fig 1C: I see no reason to cut the y-axis

Line 189: ‘This dMyc can activate…’ Remove word ‘This’

Figure 3 B and C – statistical significance

Chapter ‘miR-277 inhibits the expression of imd and Tab2-Ra/b’ – Authors should –shortly introduce why they are interested in Tab2, i.e. that it has been shown to be a key component of the Drosophila Imd pathway (Kleino et al., 2005 EMBO J)

Fig 7C – there is a mistake in the y-axis (spelling)

Line 326 - E.col infection

Figure 8 – there is no reason to cut the x-axis

Line 451: ‘And that dMyc…’ Rephrase

**Have all data underlying the figures and results presented in the manuscript been provided?**

Reviewer #1: Yes

Reviewer #2: None

Reviewer #3: Yes

PLOS authors have the option to publish the peer review history of their article (what does this mean?). If published, this will include your full peer review and any attached files.

Reviewer #1: No

Reviewer #2: No

Reviewer #3: No

---

## [Decision Letter · Decision Letter 1]

13 Jul 2020

Dear Dr Ma,

We are pleased to inform you that your manuscript entitled "Drosophila Myc restores immune homeostasis of Imd pathway via activating miR-277 to inhibit imd/Tab2" has been editorially accepted for publication in PLOS Genetics. Congratulations!

Yours sincerely,

Gregory P. Copenhaver

Editor-in-Chief

PLOS Genetics

Comments from the reviewers (if applicable):

Reviewer's Responses to Questions

**Comments to the Authors:**

Reviewer #2: All questions are well answered or explained in the revision. I recommend accpetance and publication of this manuscript.

Reviewer #3: May concerns are sufficiently addressed and I would favour accepting this manuscript

**Have all data underlying the figures and results presented in the manuscript been provided?**

Reviewer #2: None

Reviewer #3: Yes

PLOS authors have the option to publish the peer review history of their article (what does this mean?). If published, this will include your full peer review and any attached files.

Reviewer #2: No

Reviewer #3: No

**Data Deposition**

http://datadryad.org/submit?journalID=pgenetics&manu=PGENETICS-D-19-02099R1

**Press Queries**

---

## [Editor Report · Acceptance letter]

12 Aug 2020

PGENETICS-D-19-02099R1 

Drosophila Myc restores immune homeostasis of Imd pathway via activating miR-277 to inhibit imd/Tab2 

Dear Dr Ma, 

We are pleased to inform you that your manuscript entitled "Drosophila Myc restores immune homeostasis of Imd pathway via activating miR-277 to inhibit imd/Tab2" has been formally accepted for publication in PLOS Genetics! Your manuscript is now with our production department and you will be notified of the publication date in due course.

With kind regards,

Kaitlin Butler

PLOS Genetics

On behalf of:
